# Heteroscedastic Gaussian Processes and Random Features: Scalable Motion Primitives with Guarantees

**Edoardo Caldarelli**[1] **Antoine Chatalic**[2] **Adrià Colomé**[1] **Lorenzo Rosasco**[2, 3, 4] **Carme Torras**[1]

[1]Institut de Robòtica i Informàtica Industrial, CSIC – UPC, Barcelona, Spain
[2]MaLGa Center – DIBRIS – Università di Genova, Genoa, Italy
[3]CBMM – Massachusets Institute of Technology, Cambridge, MA, USA
[4]Istituto Italiano di Tecnologia, Genoa, Italy
Correspondence to: `ecaldarelli@iri.upc.edu`

**Abstract:** Heteroscedastic Gaussian processes (HGPs) are kernel-based, non-parametric models that can be used to infer nonlinear functions with time-varying noise. In robotics, they can be employed for learning from demonstration as motion primitives, i.e. as a model of the trajectories to be executed by the robot. HGPs provide variance estimates around the reference signal modeling the trajectory, capturing both the predictive uncertainty and the motion variability. However, similarly to standard Gaussian processes they suffer from a cubic complexity in the number of training points, due to the inversion of the kernel matrix. The uncertainty can be leveraged for more complex learning tasks, such as inferring the variable impedance profile required from a robotic manipulator. However, suitable approximations are needed to make HGPs scalable, at the price of potentially worsening the posterior mean and variance profiles. Motivated by these observations, we study the combination of HGPs and random features, which are a popular, data-independent approximation strategy of kernel functions. In a theoretical analysis, we provide novel guarantees on the approximation error of the HGP posterior due to random features. Moreover, we validate this scalable motion primitive on real robot data, related to the problem of variable impedance learning. In this way, we show that random features offer a viable and theoretically sound alternative for speeding up the trajectory processing, without sacrificing accuracy.

**Keywords:** Gaussian process regression, random features, motion primitives.

## 1 Introduction

*Learning from demonstration* (LfD) is a broadly used technique to transfer skills from humans to robots in a flexible and intuitive way [1]. Within the context of robotics manipulation, LfD consists of recording the movement of an arm performing a specific task multiple times. This data is then used to fit a model of the trajectory, called a *motion primitive* in this context, which allows the robot to reproduce the skill of interest. One popular way to achieve this objective is by means of the so-called *Gaussian process* (GP) regression [2]. Being a Bayesian model, a GP naturally provides a time-varying reference signal to be followed by the robot (the *GP posterior mean*), and an uncertainty quantification (the *GP posterior variance*) around the reference signal.

When used for LfD, GPs are usually *heteroscedastic* (HGP), i.e., the variance of the noise corrupting the recorded trajectories is not constant [3, 4]. As discussed by Arduengo et al. [5], such a time-varying noise variance is a key asset for modeling motion variability when HGPs are used as motion primitives. The inconsistency of the human demonstrations, captured by the time-dependent noise variance, is a form of epistemic uncertainty that is intrinsic to the task, i.e., it cannot be resolved by increasing the number of human demonstrations. Moreover, the HGP posterior variance quantifies the input-dependent uncertainty on the predictions, which is relevant when the testing points are far from the training data due to gaps in the human demonstrations.

7th Conference on Robot Learning (CoRL 2023), Atlanta, USA.

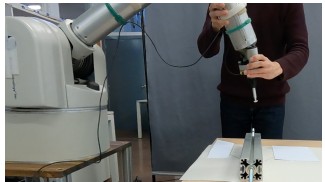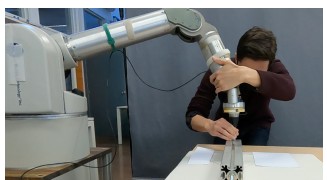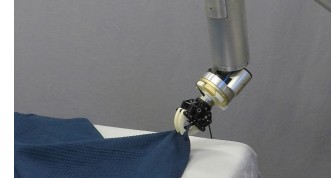

Figure 1: The tasks involved in our experiments. In the first two pictures, the human is guiding the robot to perform: a free motion above a metal piece; the insertion of the end-effector in the trail inside the piece. In the last picture, the robot is pulling a bed sheet to remove the wrinkles.

While being attractive for the aforementioned reasons, GP regression suffers from a cubic scaling in the number of training samples as it requires an inversion of the kernel matrix. For this reason, several approximation techniques have been deployed, that allow for a *linear* scaling in the number of observations. In particular, these approximations can be grouped into variational ones and the spectral ones. The former methods are data-dependent and rely on a variational approximation of the posterior distribution of the GP [6, 7, 8]. Spectral approximations, on the other hand, are most often data-independent, and can rely, e.g., on polynomial approximations [9, 10], a truncation of the kernel's spectrum [11], or a randomized approximation of an integral representation of the kernel function. The latter technique includes the well known *random features* (RFs) [12], that have successfully been used for GPs approximation [13, 14, 15, 16, 17]. In order to achieve a satisfactory model of the trajectory of interest, these approximations should not deteriorate the posterior mean and variance as these quantities fully specify the robot's motion; when such guarantees can be rigorously obtained, approximate GPs become an appealing motion primitive [11, 18, 19].

**Contributions**   In this work, we study the combination of vector-valued HGPs with RFs for fast trajectory processing in LfD. In order to assess the reliability of the approximation scheme, we perform a theoretical analysis and propose novel bounds on the approximation error of the posterior mean and variance of the HGP. This is achieved by leveraging the deep connection between approximate GP regression and *kernel ridge regression* with RFs [20]. Moreover, given the well-studied relationship between the HGP posterior variance and the time-dependent robot stiffness employed in *variable impedance control* (VIC) [21, 22, 23, 24], we further assess the quality of the approximation for the three VIC tasks described by Caldarelli et al. [24], and shown in Figure 1. Overall, we demonstrate that the combination of RFs and HGPs is theoretically sound, and significantly improves the running time of the motion primitive fitting, without sacrificing accuracy.

## 2   Related Work

**Motion primitives**   One of the most popular trajectory representations appearing in literature is given by the so-called *probabilistic movement primitives* (ProMPs) [25, 26, 27]. ProMPs model the trajectory at each time-step as a parametric weighted combination of basis functions, and are therefore different from HGPs, which are non-parametric by definition. ProMPs can be connected with another class of trajectory models, namely *dynamic movement primitves*, as shown by Li et al. [28]. On the other hand, GP-based models exhibit strong analogies with the *kernelized movement primitive* model introduced by Huang et al. [29]. In general, these approaches differ from GP regression as they also capture correlation between the different degrees of freedom (DOFs) of the trajectory. However, GPs can also be extended to account for output correlations by leveraging suitable coregionalization methods [30], based, e.g., on Gaussian mixture models [31].

**Spectral approximations**   Some theoretical properties of spectral approximations for homoscedastic, scalar-valued GPs have been explored in previous works. Särkkä and Piché [9] propose a feature approximation based on different polynomial approximations of the RBF kernel function. They are able to prove uniform convergence of the kernel, mean and variance, but without convergence rates. Moreover, Solin and Särkkä [10] propose to compute the features based on the Fourier transform of the Laplace operator. They prove convergence of the kernel function values and convergence of the posterior mean and variance, but with a dependency on the size of the domain of the

expansion of the Laplace operator. RFs are also widely used in the Bayesian optimization setting, as they allow to provide scalable algorithms with theoretical guarantees [11, 19]. In this scenario, the *uniform convergence* of the approximate kernel function is of interest as the optimization algorithms may require sampling points in the whole domain of the function being optimized. Compared to our work, as we will discuss in Section 4, these bounds have a worse dependency on the number $n$ of training samples (which is not due to the uniformity of the bounds). Although approximate GPs are known to suffer from the so-called *variance starvation* phenomenon, leading to poor estimates of the posterior variance when moving far from the training data [32], we stress that this is not problematic in LfD where one is mainly interested in the interpolation of densely sampled trajectories, whose boundaries are delimited by the task duration.

**Variational methods** Multiple methods based on variational approximations of the posterior distribution of the GP have been proposed. They typically rely on the choice of *inducing points*, i.e. of a subset of the data summarizing the whole training set [6, 7, 8]. One fundamental convergence result for the Kullback-Leibler (KL) divergence between the variational and exact GP posterior was proven by Burt et al. [18]. Moreover, Burt et al. [33, Proposition 1] showed that the error on the KL divergence implies convergence of the variational posterior means and variances. However, their bound depends on the value of the exact posterior variance, which makes the comparison with our result difficult, as it will be clear from Section 4. Hence, recent approaches such as variational Fourier features (VFFs) by Hensman et al. [8], which have shown good performance in practice, especially in the case of billions of datapoints, do not enjoy strong theoretical guarantees as RFs. Moreover, VFFs are limited to the Matérn class of kernels (and the work by Dutordoir et al. [34] provides an extension to stationary kernels on the sphere), while RFs can be applied to any stationary kernel.

## 3 Background: Approximating HPGs with Random Features

**HGP regression for trajectory encoding** Let $\mathcal{X}$ be an input space, and $x : \mathcal{X} \to \mathbb{R}$ be a scalar-valued function. In LfD, the function $x$ depends on time, i.e., $\mathcal{X} = \mathbb{R}_{\geq 0}$. A GP [2] specifies a prior distribution over the function $x$, which depends on a mean function $\mu : \mathcal{X} \to \mathbb{R}$ and a kernel $k : \mathcal{X} \times \mathcal{X} \to \mathbb{R}$. We say that $x$ follows a GP with mean $\boldsymbol{\mu}$ and kernel function $k$ if for any vector of time-steps $\mathbf{t} \in \mathbb{R}^n$, the vector $\mathbf{x}(t)$ of evaluations of $x$ at $\mathbf{t}$ follows the multivariate normal distribution $\mathbf{x}(\mathbf{t}) \sim \mathcal{N}(\boldsymbol{\mu}(\mathbf{t}), K)$, where $K \in \mathbb{R}^{n \times n}$ is defined as $K_{i,j} = k(t_i, t_j)$. The mean function is usually assumed to be 0, as the function values are standardized. The kernel function depends on a set of hyperparameters, $\boldsymbol{\theta}$, which can be fixed or inferred from data. In LfD, the time-dependent trajectory to be encoded consists of $d$ DOFs. If all DOFs are fully observed, each of them is usually modeled by an independent GP [35]. This type of estimation can be linked to a reproducing kernel Hilbert space (RKHS) of vector-valued functions [30]. We assume to have access to $\mathbf{y} \in \mathbb{R}^n$, a vector of noisy measurements of $\mathbf{x}(\mathbf{t})$. As shown by Arduengo et al. [35], assuming a constant noise variance in LfD might severely limit the quality of the posterior prediction. Therefore, the noise is time-dependent, turning the GP into an HGP. The posterior distribution of an HGP, conditioned on the noise variance values at the training points and the function's observations, is analytically available. Let $t^*$ be a testing point, and $\mathbf{k}_{t^*} := [k(t_1, t^*), \ldots, k(t_n, t^*)]^T \in \mathbb{R}^n$. Lastly, let $\Sigma_{\text{noise}} \in \mathbb{R}^{n \times n}$ be the noise variance at the training points, and $\sigma^2_{\text{noise}, t^*}$ be the noise variance at the testing point. The posterior mean and variance of the associated HGP are, respectively, [35]

$$\mu_{\text{post}}(t^*) = \mathbf{k}_{t^*}^T (K + \Sigma_{\text{noise}})^{-1} \mathbf{y}, \tag{1}$$

$$\sigma^2_{\text{post}}(t^*) = k(t^*, t^*) + \sigma^2_{\text{noise}, t^*} - \mathbf{k}_{t^*}^T (K + \Sigma_{\text{noise}})^{-1} \mathbf{k}_{t^*}. \tag{2}$$

**Random features** Let $(\Omega, \mathcal{A}, \pi)$ be a probability space over the sample space $\Omega$, and let $\psi : \Omega \times \mathcal{X} \to \mathbb{R}$. Random features (RFs) are a class of randomized methods that can be used to approximate a kernel function admitting an integral representation of the form

$$k(t, t') = \int_\Omega \psi(\omega, t) \psi(\omega, t') d\pi(\omega) \tag{3}$$

by dicretizing it. This is the case for standard kernels such as the ones belonging to the Matérn family or the radial-basis-function (RBF) kernel, as discussed by Rahimi and Recht

[12]. The expectation above is approximated by sampling $m$ vectors $(\omega_j)_{1 \leq j \leq m} \sim \pi(\omega)$. For $\tilde{\phi}(t) := \frac{1}{\sqrt{m}} [\psi(\omega_1, t), \ldots, \psi(\omega_m, t)]^T$, the kernel is approximated as $\tilde{k}(t, t') := \tilde{\phi}(t)^T \tilde{\phi}(t')$. For instance for the case of a positive definite stationary kernel $k(t, t') = k(t - t')$, Bochner's theorem ensures that $k$ has a non-negative Fourier transform, which can thus be used in place of $\pi$ while defining $\psi$ to be a trigonometric function [12]. Rahimi and Recht [12] show that, for the RBF kernel $k(t, t') = \sigma_{RBF}^2 \exp\left(-\|t - t'\|^2/(2l^2)\right)$, the values of $\omega$ are s.t. $\omega_i \sim (2\pi l^{-2})^{-\frac{m}{2}} \exp\left(-l^2 |\omega|^2/2\right)$. Then, for $b$ uniformly sampled from $[0, 2\pi)$, they define $\psi(\omega, t) := \sqrt{2}\sigma_{RBF} \cos(\omega t + b)$. To retrieve closed-form expressions for the posterior mean and variance of an HGP approximated with RFs (in the following referred to as RF-HGP), we can define the matrix $\tilde{K} \in \mathbb{R}^{n \times n}$ with entries $\tilde{K}_{i,j} = \tilde{k}(t_i, t_j)$. For any testing point $t^*$ we define $\tilde{\mathbf{k}}_{t^*} = [\tilde{k}(t_1, t^*), \ldots, \tilde{k}(t_n, t^*)]^T \in \mathbb{R}^n$. Lastly, let $\Sigma_{noise}$ and $\sigma_{noise,t^*}^2$ be as in the previous paragraph. The posterior mean and variance of the associated HGP are, respectively [35]

$$\mu_{post}(t^*) = \tilde{\mathbf{k}}_{t^*}^T (\tilde{K} + \Sigma_{noise})^{-1} \mathbf{y}, \tag{4}$$

$$\sigma_{post}^2(t^*) = \tilde{k}(t^*, t^*) + \sigma_{noise,t^*}^2 - \tilde{\mathbf{k}}_{t^*}^T (\tilde{K} + \Sigma_{noise})^{-1} \tilde{\mathbf{k}}_{t^*}. \tag{5}$$

We can observe that these equations are structurally the same as Equations (1) and (2), with the approximated kernel $\tilde{k}$ replacing the exact kernel $k$. Fixing the dimensionality of the RF vector, RF-HGPs allow for linear complexity in the number of samples when computing the posterior distribution of the HGP, as the posterior expressions in Equations (4) and (5) can be simplified by the well-known *Woodbury identity* [36]. Such an identity allows to perform matrix inversion in $\mathcal{O}(m^3)$, $m \ll n$ being the dimension of the RF vector, as shown in Appendix A. Moreover, the kernel approximation is data-independent, as the functions $\psi(\omega_j, \cdot)$ can be computed prior to seeing any data.

## 4 Quantification of the Approximation Errors

We now analyse the proposed approximate HGPs in two different settings. In the first one, the hyperparameters $\boldsymbol{\theta}$ of the kernel function (which typically contains at least a lengthscale parameter), as well as the noise variance values at the training points, $\Sigma_{noise}$, and at a testing point, $\sigma_{noise,t^*}^2$, are fixed *a priori* (**oracle setup**). In the second setting, $\boldsymbol{\theta}$, $\Sigma_{noise}$, and $\sigma_{noise,t^*}^2$ are directly inferred from data (**heuristic setup**). In the first case, the error introduced by the RF approximation can be rigorously quantified, as we will show in the remainder of this section. In the second case, convergence to the exact HGP can be attained empirically, and a suitable inference algorithm needs to be used, as we will discuss in Section 5. Both in the oracle and in the heuristic setup, the RF-HGP's posterior distribution is computed at a set of testing time-steps $\mathcal{T} = \{t_1^*, \ldots, t_T^*\}$, which are fixed *a priori* based on the desired task duration. In this section, we derive error bounds on posterior mean and variance for the RF-HGP, at the test points in $\mathcal{T}$, within an oracle setup. These results guarantee that RFs are a suitable approximation strategy, that can be used in robotics scenarios. The proofs, reported in Appendix B.2 and Appendix B.3, rely on techniques developed by Rudi and Rosasco [20].

### 4.1 Assumptions

In the following, we set $\psi_j(\cdot) := \psi(\omega_j, \cdot)$, where the latter function is defined in Section 3. Moreover, we make the following assumptions.

**Assumption 4.1.** The kernel is bounded, i.e., $k(t, t') \leq \kappa$. Moreover, the kernel admits the integral representation of Equation (3), in terms of a suitable function $\psi(\omega, \cdot)$.

**Assumption 4.2.** The entries of the RF vector are bounded, i.e., $|\psi_j(t)| \leq \alpha, \forall j \in \{1, \ldots, m\}, \forall t$.

**Assumption 4.3.** The noise variance values in the diagonal matrix $\Sigma_{noise}$ are bounded from below, i.e. $\exists \gamma \in (0, 1]$ s.t. $(\Sigma_{noise})_{ii} \geq \gamma n, \forall i \in \{1, \ldots, n\}$.

Assumption 4.1 is standard, and holds, e.g., for the RBF kernel ($\kappa = \sigma_{RBF}^2$). Assumption 4.2 holds, e.g., for the RFFs described in Section 3 (with $\alpha = \sqrt{2}\sigma_{RBF}$). Assumption 4.3 is obtained by replacing, e.g, $K + \Sigma_{noise}$ in Equations (1) and (2) with $K + (\Sigma_{noise} + \gamma n I)$, a common practice aimed at avoiding numerical instabilities in matrix inversion. Lastly, for the RBF kernel and RFFs, $\kappa = \alpha^2$.

### 4.2 Concentration of Posterior Mean and Variance

Under these assumptions, we can derive the two main results of this paper, bounding the deviation of the posterior mean and variance of our RF-HGP w.r.t. its exact counterpart.

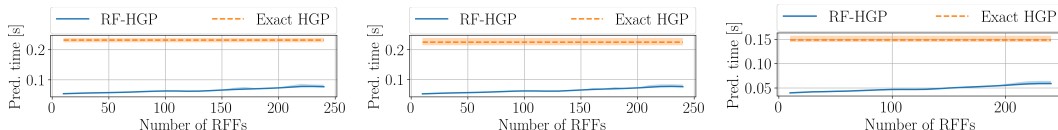

Figure 2: Time taken to compute the posterior distribution with an exact HGP and an RF-HGP, for free motion (left), assembly (center) and bed-making (right) tasks, in an oracle setup. Median, the $15^{th}$ and $85^{th}$ percentiles across all DOFs, 50 seeds.

**Theorem 4.4.** *Let* $\delta \in (0,1]$, *and* $m$ *be the dimension of the RF vector. Let* $m \geq 8\left(\frac{1}{3} + \frac{\alpha^2}{\gamma}\right)\log(\frac{8\alpha^2}{\gamma\delta})$, *and consider a vector-valued HGP with* $d$ *independent components. Let* $\boldsymbol{\mu}_{post}$ *and* $\tilde{\boldsymbol{\mu}}_{post}$ *denote its exact and RF-approximated posterior mean as of Equations* (1) *and* (4). *Let* $\nu = \max_{1\leq i\leq d}\left\|\frac{1}{\sqrt{n}}\mathbf{y}_i\right\|$, *where* $\mathbf{y}_i$ *denotes the observations of DOF* $i$. *Lastly, let* $\mathcal{T}$ *be a set of testing points. Under Assumptions 4.1 to 4.3, with probability at least* $1 - d\delta$, $\forall t^* \in \mathcal{T}$, *it holds that*

$$\|\boldsymbol{\mu}_{post}(t^*) - \tilde{\boldsymbol{\mu}}_{post}(t^*)\|_2 \leq \nu\alpha^2\sqrt{\frac{2d\log\frac{2|\mathcal{T}|n}{\delta}}{m\gamma^2}} + \frac{\sqrt{2d}\kappa\nu}{\sqrt{\gamma}}\left[\frac{2\log\frac{8\kappa^2}{\gamma\delta}(1+\alpha^2/\gamma)}{3m} + \alpha\sqrt{\frac{2\log\frac{8\kappa^2}{\gamma\delta}}{\gamma m}}\right].$$

**Theorem 4.5.** *Under the assumptions of Theorem 4.4, denoting* $\boldsymbol{\sigma}^2_{post}$ *and* $\tilde{\boldsymbol{\sigma}}^2_{post}$ *respectively the exact and RF-approximated posterior variances as of Equations* (2) *and* (5), *it holds with probability at least* $1 - d\delta$, $\forall t^* \in \mathcal{T}$,

$$\|\boldsymbol{\sigma}^2_{post}(t^*) - \tilde{\boldsymbol{\sigma}}^2_{post}(t^*)\|_2 \leq \alpha^2\sqrt{\frac{2d\log\frac{2|\mathcal{T}|}{\delta}}{m}} + \left(\frac{\kappa\alpha^2}{\sqrt{\gamma}} + \frac{\alpha^4}{\gamma}\right)\sqrt{\frac{2d\log\frac{2|\mathcal{T}|n}{\delta}}{m}}$$
$$+ \frac{\alpha^3\sqrt{2d}}{\sqrt{\gamma}}\left[\frac{2\log\frac{8\kappa^2}{\gamma\delta}(1+\alpha^2/\gamma)}{3m} + \alpha\sqrt{\frac{2\log\frac{8\kappa^2}{\gamma\delta}}{\gamma m}}\right].$$

Our bounds show that both the mean and the variance errors for a vector-valued RF-HGP scale as $\mathcal{O}(m^{-1/2})$. Our results cover the RF-based approximation of the homoscedastic GP as a special case taking $\Sigma_{\text{noise}} = \beta I$ for some $\beta \geq 0$. Our results ensure that RFs are a viable data-independent approximation method, and match the rates of other data-dependent strategies, such as the well-studied *Nyström method* [37] for GPs, as shown by Lu et al. [38]. Furthermore, by inspecting the proofs, and in particular the results in Appendix C, one can observe that the assumptions of the LfD framework (fixed testing points, scalar domain) translate in a dependency of the bounds on $|\mathcal{T}|$. Nonetheless, our proofs can easily be adapted to the case of dense, multidimensional domains by replacing Corollary C.2 with the uniform convergence result for RFs [39, 40] (rather than using a point-wise bound and a union bound), which would leave the overall error in $\mathcal{O}(m^{-1/2})$.

As mentioned in Section 2, our bounds strictly improve over the state-of-the art bounds by Mutny and Krause [11]. For instance, Mutny and Krause [11, Proof of Theorem 5 and Proposition 1] in the context of homoscedastic GP approximated with deterministic features provide bounds of the form $\sup_{t^*}|\mu_{\text{post}}(t^*) - \tilde{\mu}_{\text{post}}(t^*)| \lesssim \epsilon\nu n^2\sigma^{-2}$ and $\sup_{t^*}|\sigma^2_{\text{post}}(t^*) - \tilde{\sigma}^2_{\text{post}}(t^*)| \lesssim \epsilon n^3\sigma^{-2}$ where $\epsilon$ denotes the accuracy of a uniform bound on the kernel approximation, i.e. $\sup_{x,y}|k(x,y) - \tilde{k}(x,y)| \leq \epsilon$. In our setting, when $d = 1$ and in the homoscedatic setting $\Sigma_{\text{noise}} = \sigma^2 I$ and $\gamma = \sigma^2/n$ (cf. Assumption 4.3) we obtain a dependence in $n$ of order $\mathcal{O}(n^{3/2}\log n)$ both for the posterior mean and variance which strictly improves on these bounds. This result can be derived, e.g., for the mean by observing that the rate with respect to $\gamma$ translates to an error upper bound with complexity $O(n\sqrt{\log n}) + O(n^{3/2}\log n) + O(n\sqrt{\log n})$.

## 5 Empirical Evaluation

In this section, we empirically validate the RF-HGP. The implementation is built on the package `gpflow` by Matthews et al. [41][1]. We choose RFFs to approximate the RBF kernel, as reported

---

[1]The open-source code for the experiments is available at `https://github.com/LCSL/rff-hgp`.

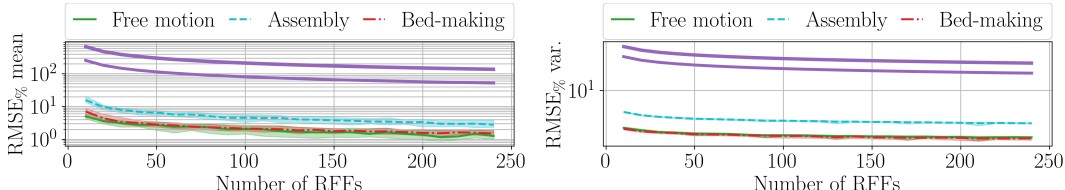

Figure 3: RMSE between the posterior means (left) and posterior variances (right) of an HGP and an RF-HGP, in an oracle setup, for three different tasks. Median, $15^{th}$ and $85^{th}$ percentiles across all DOFs, 50 seeds. The purple curves are the theoretical rates including the dependency on $\nu$, $\gamma$ and $m$, and show that the overall rate in $1/\sqrt{m}$ matches experimental result. The lowest purple curve is for the bed making task, while the rates for assembly and proof-of-concept coincide.

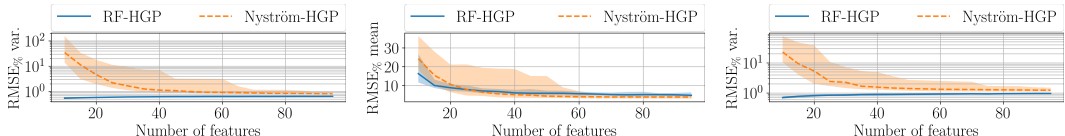

Figure 4: Incremental learning with missing chunks of data, in an oracle setup, for free motion (left), assembly (center) and bed-making (right) tasks. RF-HGP vs a Nyström approximation with fixed centers, to achieve the same complexity as RFs. Median, $15^{th}$ and $85^{th}$ percentiles across all DOFs, 50 seeds.

in Section 3. In this work, we do not train the RFF parameters $\omega$'s, as it is prone to overfitting [13, 42], and annuls the Monte-Carlo interpretation of RFFs introduced in Section 3. To assess the correctness of the GP approximation, we process real demonstrations of different robotic tasks, obtained by means of kinesthetic teaching with a 7-DOF Barrett WAM manipulator. The trajectories are recorded while performing a movement in free Cartesian space, an assembly task, and a bed-making skill [24]. These trajectories are particularly interesting from the VIC point of view, and summarized in Figure 1. Since the time-dependent impedance, or stiffness, of a manipulator can be tuned based on the variance of the human demonstrations [21, 23, 22, 24], it is important that the RF approximation does not deteriorate the quality of the posterior distribution of the HGP. The first task exhibits varying boundaries in which the manipulator can move, while the latter two involve physical constraints on the robot's motion. The assembly task requires a motion with low stiffness when the pieces being mounted are in contact. On the other hand, the bed-making skill requires the robot to stiffen up to remove wrinkles from the sheet's surface, in spite of it opposing the motion of the robot's end-effector. Each task uses a different number of human demonstrations, namely 6, 7 and 5. While the number of demonstration is relatively small, the total number of training points is 1286 for the proof-of-concept experiment, 1222 for the assembly task, and 864 for the bed-making skill, justifying the need for scalable GPs. Moreover, the number of samples might increase for longer or denser trajectories. We consider an oracle setup in Section 5.1, where all hyperparameters and noise variances are assumed to be known, and then discuss and evaluate in Section 5.2 the heuristic case of unknown hyperparameters and noise variance.

## 5.1   Oracle Setup

As a start, we consider the case in which the GP kernel hyperparameters $\boldsymbol{\theta}$ are known, and the noise variance values $\Sigma_{\text{noise}}$, $\sigma^2_{\text{noise},t^*}$ are provided by an oracle, as discussed in Section 4. This experiment is useful to assess how the prediction capability of RF-HGP deteriorates due to the kernel spectral approximation, in view of the theoretical analysis carried out in Section 4. The oracle is given by an exact HGP trained with the EM algorithm by Kersting et al. [4], for 15 iterations per DOF.

**Offline learning**   In this setup, we are interested in comparing the prediction times (Figure 2) and accuracy (Figure 3) of exact and approximated GP regression. By prediction time, we refer here to the time taken to compute the posterior means and posterior variances at the testing points. The human demonstrations are temporally aligned on the same interval, and processed all together, in an offline fashion. The prediction times, as shown in Figure 2, indicate that the RF-HGP is more

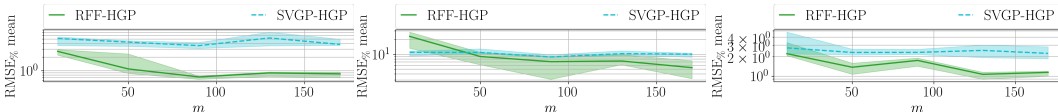

Figure 5: RMSE between the posterior means of an exact and an approximated HGP, in heuristic setup, for free motion (left), assembly (center) and bed-making (right) tasks. Median, $15^{th}$ and $85^{th}$ percentiles, across all DOFs and 5 seeds.

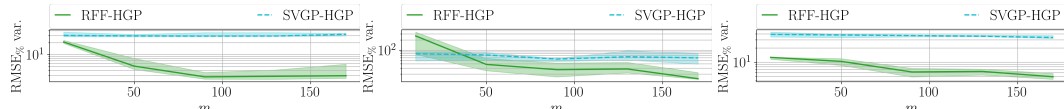

Figure 6: RMSE between the posterior variances of an exact and an approximated HGP, in heuristic setup, for free motion (left), assembly (center) and bed-making (right) tasks. Median, $15^{th}$ and $85^{th}$ percentiles, across all DOFs and 5 seeds.

efficient, as expected. Moreover, to asses the predictive accuracy, we compute the normalized root-mean-squared-error (RMSE) between the posterior means or variances of the HGP and RF-HGP:

$$\text{RMSE}_{\%_\text{mean}} = \sqrt{\frac{1}{d} \sum_{i=1}^{d} \frac{\sum_{t \in \mathcal{T}} (\tilde{\mu}_{\text{post},i}(t) - \mu_{\text{post},i}(t))^2}{\sum_{t \in \mathcal{T}} \mu_{\text{post},i}(t)^2}} \cdot 100. \tag{6}$$

The $\text{RMSE}_\%$ on the variance is computed in the same way. This experiment allows us to observe that the error rate follows the expected rate in $\mathcal{O}(m^{-1/2})$, as shown in Figure 3, indicating that RFs can be used to speed up the inference without sacrificing accuracy in the posterior's calculation. We can also observe that although all errors decrease with an increasing number of features, the assembly task exhibits the largest error among the three tasks. This is likely to be due to the RF approximation being challenged by the sharp changes in the task variability.

**Incremental learning**    RFs are data-independent, and they can be used to perform incremental learning of the posterior distribution of the HGP [43]. In an oracle setup, the posterior distribution can be updated every time a new demonstration is gathered (see Appendix D). This approach is also possible, e.g., with the Nyström approximation of the kernel function, provided that the Nyström centers are sampled from the first demonstration and not updated afterwards. Doing so would imply re-computing the correlation matrices between the centers and the whole set of training points, which has a linear complexity in the size of the training set. Keeping a fixed set of Nyström centers is not an issue in general for LfD, as the domain is scalar and all demonstrations are temporally aligned on the time interval $[0, 1]$, and thus the first demonstration already covers the whole domain. However, this fact does not hold if, for any reason, portions of the human demonstrations are missing or invalid. In this case, the initialization of the centers might be poor, and the posterior calculation spoiled. This issue is shown in the examples of Figure 4, where chunks of 60 observations, sampled uniformly at random, were removed from each demonstration. RFFs offer a more reliable option, in such a scenario.

## 5.2    Heuristic Setup

After validating our theoretical results, we now consider the more realistic setting in which the kernel hyperparameters and noise variance are unknown and need to be heuristically estimated from data. Note that this step can serve as preliminary stage with a few demonstrations before applying, e.g., the incremental learning described in the previous section. Here, we train the RF-HGP as follows.

**Workflow with hyperparameters tuning**    Here, we consider the expectation-maximization (EM) training proposed by Kersting et al. [4], adapted to the RF-HGP. The algorithm comprises the following steps, which are performed independently for each DOF (all the GPs involved take time as input):

1. train a first homoscedastic GP ($\mathcal{GP}_1$), approximated with RFs (RF-GP), on the demonstrations' data with a maximum likelihood estimate (MLE), to retrieve $\boldsymbol{\theta}$;

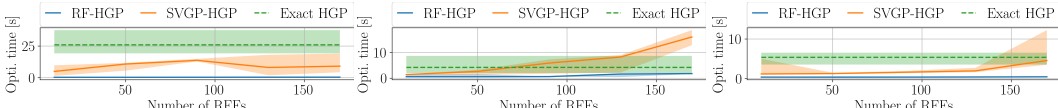

Figure 7: Time taken to perform an MLE with an exact HGP and an RF-HGP, with demonstrations of free motion (left), assembly (center) and bed-making (right) tasks. Median, the $15^{th}$ and $85^{th}$ percentiles, across all DOFs and 5 seeds.

2. compute the mean-squared-error (MSE) between the posterior mean of $\mathcal{GP}_1$ and the demonstrations' data, at each time-step;
3. train a second RF-GP ($\mathcal{GP}_2$) by means of MLE, using the MSE as training data; $\mathcal{GP}_2$ is a surrogate model of the time-dependent noise variance function;
4. compute the posterior mean and variance of a new RF-HGP ($\mathcal{GP}_3$) as in Equations (4) and (5), with the kernel hyperparameters $\boldsymbol{\theta}$ from step 1 and the noise variance values in $\Sigma_{\text{noise}}$ and $\sigma^2_{\text{noise},t^*}$ given by the posterior mean of $\mathcal{GP}_2$, evaluated at the training and testing time-steps respectively;
5. compute MSE between the posterior mean of $\mathcal{GP}_3$ and the demonstrations, at each time-step;
6. repeat from step 3 until convergence or until the maximum number of iterations is reached.

Sampling first whole noise variance profiles from the posterior of $\mathcal{GP}_2$, and then whole trajectories from the posterior specified by Equations (1) and (2) conditioned on the noise variance, yields trajectories that are strongly correlated in the regions of low variance, and vice-versa.

**Results**    Figure 7 reports the time taken to complete an MLE for retrieving the GP hyperparameters, both in the exact and in the RFF setup. Considering the errors reported in Figures 5 and 6, we observe that convergence to the exact HGP posterior can be heuristically attained in the worst possible scenario of having no knowledge about the necessary GP priors. Again, the largest errors are attained by the assembly task, as discussed for the oracle setup. Concerning variational methods, such as the sparse variational GP (SVGP) by Hensman et al. [44], Figure 7 shows that SVGP, employed in the EM training algorithm, requires many more iterations per step to converge, in the hyperparameter training, due to the greater complexity of the optimization problem being solved. To overcome this issue, we set the maximum number of iterations per step to 100. However, as we can observe from Figures 5 and 6, this choice hinders the convergence through the EM training process, and our proposed feature-based approach has a strictly better accuracy-computational complexity trade-off.

## 6   Limitations

The assumptions behind our theoretical analysis were stated in Section 4.1. Moreover, as discussed in Section 4, our theoretical analysis focuses on an oracle setup. This type of scenario is standard in kernel theory [45, 20, 38], as it allows to decouple the error due to the kernel approximation from the uncertainty surrounding the GP hyperparameters. If this oracle setup does not hold, the HGP training method is heuristic, and uses two approximate GPs iteratively in an EM fashion. The errors displayed in the heuristic experiments of Section 5.2 may change with a different training algorithm.

## 7   Conclusion

In this work, we have studied the combination of heteroscedastic Gaussian processes and random features, used as scalable motion primitives in the context of learning from demonstration. In a theoretical analysis, we derived novel upper bounds on the approximation error, induced by random features, on the posterior mean and variance of a Gaussian process. Moreover, we have validated this approximate motion primitive w.r.t. relevant tasks for variable impedance control of robotic manipulators, namely, a motion in free Cartesian space, an assembly task, and a bed-making skill. Our theoretical and empirical results demonstrate that random features are a theoretically sound approximation method, that can be used to speed up the motion primitive fitting without sacrificing accuracy. Moreover, we have shown that random features are well suited to incremental learning from demonstration, thanks to their data-independent nature.

**Acknowledgments**

E. Caldarelli, A. Colomé and C. Torras acknowledge support from the project CLOTHILDE ("CLOTH manIpulation Learning from DEmonstrations"), funded by the European Research Council (ERC) under the European Union's Horizon 2020 research and innovation programme (Advanced Grant agreement No 741930). E. Caldarelli acknowledges travel support from ELISE (GA No 951847). L. Rosasco acknowledges the financial support of the European Research Council (grant SLING 819789), the AFOSR projects FA9550-18-1-7009, FA9550-17-1-0390 and BAA-AFRL-AFOSR-2016-0007 (European Office of Aerospace Research and Development), the EU H2020-MSCA-RISE project NoMADS - DLV-777826, and the Center for Brains, Minds and Machines (CBMM), funded by NSF STC award CCF-1231216.

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

Table 1: Summary of notations.

| Variable | Meaning |
| --- | --- |
| $n$ | Number of training samples |
| $T$ | Number of testing samples |
| $d$ | Degrees of freedom (one HGP for each) |
| $\alpha$ | $|\psi_j(t)| \leq \alpha$ (cf. Assumption 4.2) |
| $\kappa$ | $k(t, t') \leq \kappa$ (cf. Assumption 4.1) |
| $\nu$ | $\nu = \max_{1 \leq j \leq d} \|\frac{1}{\sqrt{n}} \mathbf{y}_j\|$ |
| $\Sigma_{\text{noise}} \in \mathbb{R}^{n \times n}$ | Diagonal time-varying noise variance at training points |
| $R = \Sigma_{\text{noise}}/n$ | Normalized noise variance at training points |
| $\sigma^2_{\text{noise},t^*} \in \mathbb{R}$ | Noise variance at testing point $t^*$ |
| $r^2_{t^*} = \sigma^2_{\text{noise},t^*}/n$ | Normalized noise variance at testing point $t^*$ |
| $0 < \gamma < 1$ | $R_{ii} > \gamma, 1 \leq i \leq n$ |
| $S : \mathcal{H} \to \mathbb{R}^n$ | Sampling operator with normalization $n^{-1/2}$ |
| $L = SS^* = K/n \in \mathbb{R}^{n \times n}$ | Normalized Gram matrix with exact kernel |
| $L_R \in \mathbb{R}^{n \times n}$ | $L_R = L + R$ |
| $\psi_j(t) = \psi(\omega_j, t)$ | Element of approximate feat. vector |
| $\tilde{\phi}(t) = m^{-1/2}[\psi_1(t), \ldots, \psi_m(t)]^T$ | Approximate feat. vector |
| $S_m : \mathbb{R}^m \to \mathbb{R}^n$ | Sampling operator with normalization $n^{-1/2}$ |
| $L_m = S_m S_m^* = \tilde{K}/n \in \mathbb{R}^{n \times n}$ | Normalized Gram matrix with RF kernel |
| $L_{m,R} \in \mathbb{R}^{n \times n}$ | $L_{m,R} = L_m + R$ |

## A HGP Posterior Equations Revisited

In this section, we will rewrite the exact and approximated HGP posterior equations from Section 3 in terms of standard linear operators used in RKHS theory. For a linear operator $A$, we denote its adjoint by $A^*$. Let $\mathcal{H}$ be the RKHS associated to the kernel of interest. In order to retrieve a suitable expression, we denote $S : \mathcal{H} \to \mathbb{R}^n$ the sampling operator defined as $Sf := \frac{1}{\sqrt{n}}[f(t_1), \ldots f(t_n)]^T$. Moreover, the adjoint of the sampling operator is defined as $S^* : \mathbb{R}^n \to \mathcal{H} : S^*\mathbf{a} = \frac{1}{\sqrt{n}} \sum_{i=1}^n a_i k(t_i, \cdot)$, $a_i$ being the $i$-th entry of $\mathbf{a}$. Now, let $L : \mathbb{R}^n \to \mathbb{R}^n, L := SS^*$. Note that $K = nL$. Let $R = \frac{1}{n}\Sigma_{\text{noise}}$, let $r_{t^*} = \frac{1}{n}\sigma^2_{\text{noise},t^*}$. Lastly, let $\langle \cdot, \cdot \rangle_{\mathbb{R}^n}$ denote the inner product of $n$-dimensional vectors. With this notation, let us consider a single DOF of the trajectory to be processed. The posterior mean of the associated exact HGP from Equation (1) is

$$\mu_{\text{post}}(t^*) = \left\langle (L + R)^{-1} S k(t^*, \cdot), \frac{1}{\sqrt{n}} \mathbf{y} \right\rangle_{\mathbb{R}^n}. \tag{7}$$

Moreover, the posterior variance from Equation (2) is given by

$$\sigma^2_{\text{post}}(t^*) = k(t^*, t^*) + nr_{t^*} - \langle S k(t^*, \cdot), (L + R)^{-1} S k(t^*, \cdot) \rangle_{\mathbb{R}^n}. \tag{8}$$

Considering RFs, we can define the operator $S_m : \mathbb{R}^m \to \mathbb{R}^n, S_m := \frac{1}{\sqrt{n}}[\tilde{\phi}(t_1) \ldots, \tilde{\phi}(t_n)]^T$, and $L_m : \mathbb{R}^n \to \mathbb{R}^n, L_m := S_m S_m^*$. With this notation, let us consider a single DOF of the trajectory to be processed. The RF-based posterior mean of the associated HGP from Equation (4) can be rewritten as

$$\tilde{\mu}_{\text{post}}(t^*) = \left\langle (L_m + R)^{-1} S_m \tilde{\phi}(t^*), \frac{1}{\sqrt{n}} \mathbf{y} \right\rangle_{\mathbb{R}^n}. \tag{9}$$

On the other hand, the RF-based posterior variance from Equation (5) is given by

$$\tilde{\sigma}^2_{\text{post}}(t^*) = \tilde{k}(t^*, t^*) + nr_{t^*} - \langle S_m \tilde{\phi}(t^*), (L_m + R)^{-1} S_m \tilde{\phi}(t^*) \rangle_{\mathbb{R}^n}. \tag{10}$$

A summary of the main operators and constants that will appear in the proofs can be found in Table 1.

**Fast matrix inversion** By defintion, the operators $L_m$ and $S_m$ are matrices. The inversion of the matrix $L_m + R$ appearing in Equations (9) and (10) can be performed by means of Woodbury identity [36], as follows:

$$L_{m,R}^{-1} = (S_m S_m^* + R)^{-1} \tag{11}$$

$$= R^{-1} - R^{-1} S_m (I + S_m^* R^{-1} S_m)^{-1} S_m^* R^{-1}. \tag{12}$$

The latter expression involves inverting an $m \times m$ matrix, which boosts the speed of the HGP posterior calculation if $m \ll n$.

# B    Proofs of the Main Results

In this appendix, we report the proofs of the two main theoretical results of our paper, along with some technical propositions that will be extensively used. In the following, we denote by $A_R$ the operator $A + R$, with $R$ diagonal positive definite matrix, and by $A_\gamma$ the operator $A + \gamma I$. Moreover, in the remainder, $\|\cdot\|$ denotes the operator norm, while $\|\cdot\|_2$ denotes the Euclidean norm of a vector.

## B.1    Useful Propositions

In this part, we report three propositions that will be useful in the proofs.

**Proposition B.1** (Proposition 8 of [20]). *Let $\mathcal{H}$ be a separable Hilbert space, $A, B$ be two bounded self-adjoint positive linear operators on $\mathcal{H}$, and $\lambda > 0$. Then*

$$\|A_\lambda^{-1/2} B^{1/2}\| \le \|A_\lambda^{-1/2} B_\lambda^{1/2}\| \le \frac{1}{(1-\beta)^{1/2}}, \tag{13}$$

*where*

$$\beta = \lambda_{max}\left[ B_\lambda^{-1/2}(B - A)B_\lambda^{-1/2} \right]. \tag{14}$$

**Proposition B.2.** *Let $S_m : \mathbb{R}^m \to \mathbb{R}^n, S_m := \frac{1}{\sqrt{n}}[\tilde{\phi}(t_1) \ldots, \tilde{\phi}(t_n)]^T$, and assume that the entries of the RF vectors are bounded, that is, $|\psi_j(t)| \le \alpha, \forall j \in \{1, \ldots, m\}$. Then,*

$$\|S_m\| \le \alpha. \tag{15}$$

*Proof.* The result follows from the definition of operator norm:

$$\|S_m\| = \sup_{\mathbf{a} \in \mathbb{R}^m, \|\mathbf{a}\|_2 \le 1} \|S_m \mathbf{a}\|_2 \tag{16}$$

$$= \sup_{\mathbf{a} \in \mathbb{R}^m, \|\mathbf{a}\|_2 \le 1} \frac{1}{\sqrt{n}} \sqrt{\langle \tilde{\phi}(t_1), \mathbf{a} \rangle_2^2 + \cdots + \langle \tilde{\phi}(t_n), \mathbf{a} \rangle_2^2} \tag{17}$$

$$\le \frac{1}{\sqrt{n}} \sqrt{n\alpha^2} = \alpha, \tag{18}$$

as reported in the statement. $\qquad\square$

**Proposition B.3.** *Let $A$ be a bounded positive semi-definite operator, and let $A_R := A + R$, with $R$ diagonal positive definite and $A_\gamma = A + \gamma I$. Lastly, assume all entries in $R$ are greater or equal to $\gamma$. Then,*

$$\|A_R^{-1/2} A_\gamma^{1/2}\| \le 1. \tag{19}$$

*Proof.* Noting that $A_R - A_\gamma = (R - \gamma I) \succcurlyeq 0$ by hypothesis, it holds $A_\gamma \preccurlyeq A_R$ and thus

$$\|A_R^{-1/2} A_\gamma^{1/2}\|^2 = \|A_R^{-1/2} A_\gamma A_R^{-1/2}\| \le \|I\| = 1. \tag{20}$$

$\qquad\square$

## B.2 Proof of Theorem 4.4 (Deviation of Approximate Posterior Mean)

We report here the proof of Theorem 4.4. We start by considering a single DOF, and generalize to a $d$-valued GP at the end of this section. We begin by proving a lemma that will be used to retrieve the main result.

**Lemma B.4.** *Let* $m \geq 8 \left( \frac{1}{3} + \frac{\alpha^2}{\gamma} \right) \log(\frac{8\alpha^2}{\gamma\delta})$, *and* $\delta = (0, 1]$. *Then, the following bound holds, with probability at least* $1 - \delta$,

$$\|(L_{m,R}^{-1} - L_R^{-1})Sk(t^*, \cdot)\|_2 \leq \frac{\sqrt{2}\kappa}{\sqrt{\gamma}} \left[ \frac{2 \log \frac{8\kappa^2}{\gamma\delta}(1 + \alpha^2/\gamma)}{3m} + \sqrt{\frac{2 \log \frac{8\kappa^2}{\gamma\delta}\alpha^2}{\gamma m}} \right]. \tag{21}$$

*Proof.* In order to bound the term of interest, we can use the fact that, for any invertible matrices $A$ and $B$, $A^{-1} - B^{-1} = A^{-1}(I - AB^{-1}) = A^{-1}(B - A)B^{-1}$, and Proposition B.3, as follows:

$$\|(L_{m,R}^{-1} - L_R^{-1})Sk(t^*, \cdot)\|_2 \tag{22}$$

$$= \|L_{m,R}^{-1}(L_R - L_{m,R})L_R^{-1}Sk(t^*, \cdot)\|_2 \tag{23}$$

$$= \|L_{m,R}^{-1/2}L_{m,R}^{-1/2}L_{m,\gamma}^{1/2}L_{m,\gamma}^{-1/2}L_\gamma^{1/2}L_\gamma^{-1/2}(L - L_m)L_R^{-1}Sk(t^*, \cdot)\|_2 \tag{24}$$

$$\leq \frac{1}{\sqrt{\gamma}} \|L_{m,R}^{-1/2}L_{m,\gamma}^{1/2}\| \|L_{m,\gamma}^{-1/2}L_\gamma^{1/2}\| \|L_\gamma^{-1/2}(L - L_m)L_R^{-1}Sk(t^*, \cdot)\|_2 \tag{25}$$

$$\leq \frac{\kappa}{\sqrt{\gamma}} \|L_{m,\gamma}^{-1/2}L_\gamma^{1/2}\| \|L_\gamma^{-1/2}(L - L_m)L_\gamma^{-1/2}\| \|L_R^{-1/2}S\|. \tag{26}$$

We can now proceed to bound each of the three factors. To start off, let us consider $\|L_R^{-1/2}S\|$. This term can be bounded by using the *polar decomposition* of the bounded linear operator $S$, as follows. Let $S = (SS^*)^{1/2}U$, where $U$ is a partial isometry. By Proposition B.3, the definition of polar decomposition, and by considering that $L \preccurlyeq L_\gamma$ by definition,

$$\|L_R^{-1/2}S\| = \|L_R^{-1/2}(SS^*)^{1/2}U\| \tag{27}$$

$$\leq \|L_R^{-1/2}L^{1/2}\| \|U\| \tag{28}$$

$$\leq \|L_R^{-1/2}L_\gamma^{1/2}\| \|U\| \tag{29}$$

$$\leq 1. \tag{30}$$

Now, we can move on to bound $\|L_\gamma^{-1/2}(L - L_m)L_\gamma^{-1/2}\|$. To do so, we can observe that, by definition,

$$L_m = S_m S_m^* \tag{31}$$

$$= \frac{1}{n}\frac{1}{m} \sum_{i=1}^{m} \begin{bmatrix} \psi_i(t_1) \\ \dots \\ \psi_i(t_n) \end{bmatrix} \otimes \begin{bmatrix} \psi_i(t_1) \\ \dots \\ \psi_i(t_n) \end{bmatrix}. \tag{32}$$

Moreover, due to linearity of expectation,

$$\mathbb{E}_\omega[L_m] = L. \tag{33}$$

We can therefore apply Proposition C.4, with $p = m$, $Q = L$, and $Q_p = L_m$. Note that $\text{Tr } L$ is the trace of the normalized Gram matrix $\frac{1}{n}K$ and hence is smaller or equal to $\kappa^2$ under Assumption 4.1. Lastly, the value of the constant $\mathcal{F}_\infty(\gamma)$ in Proposition C.4 can be computed as follows:

$$\left\langle \frac{1}{\sqrt{n}} \begin{bmatrix} \psi_i(t_1) \\ \dots \\ \psi_i(t_n) \end{bmatrix}, \frac{1}{\sqrt{n}}L_\gamma^{-1} \begin{bmatrix} \psi_i(t_1) \\ \dots \\ \psi_i(t_n) \end{bmatrix} \right\rangle_{\mathbb{R}^n} \leq \frac{\alpha^2}{\gamma}. \tag{34}$$

Thus, we obtain, with probability at least $1 - \delta$,

$$\|L_\gamma^{-1/2}(L - L_m)L_\gamma^{-1/2}\| \leq \frac{2 \log \frac{8\kappa^2}{\gamma\delta}(1 + \alpha^2/\gamma)}{3m} + \sqrt{\frac{2 \log \frac{8\kappa^2}{\gamma\delta}\alpha^2}{\gamma m}}. \tag{35}$$

To conclude the proof, we can bound $\|L_{m,\gamma}^{-1/2}L_\gamma^{1/2}\|$. By Proposition B.1, we have that

$$\|L_{m,\gamma}^{-1/2}L_\gamma^{1/2}\| \leq \frac{1}{(1-\beta)^{1/2}}, \quad \text{where} \quad \beta = \lambda_{\max}\left[L_\gamma^{-1/2}(L-L_m)L_\gamma^{-1/2}\right]. \tag{36}$$

According to Equation (33), we can apply Proposition C.4 and see that with probability at least $1-\delta$

$$\beta \leq \frac{2\log\frac{8\kappa^2}{\gamma\delta}}{3m} + \sqrt{\frac{2\log\frac{8\kappa^2}{\gamma\delta}\alpha^2}{\gamma m}} \leq 0.5 \tag{37}$$

provided that $m \geq 8\left(\frac{1}{3} + \frac{\alpha^2}{\gamma}\right)\log(\frac{8\alpha^2}{\gamma\delta})$. $\qquad\square$

**Proof of Theorem 4.4** In order to retrieve the main concentration result, we can consider the following decomposition of the error on the posterior mean. By Cauchy-Schwarz inequality and Equations (7) and (9),

$$|\tilde{\mu}_{\text{post}}(t^*) - \mu_{\text{post}}(t^*)| = \left|\left\langle L_{m,R}^{-1}S_m\tilde{\phi}(t^*) - L_R^{-1}Sk(t^*,\cdot), \frac{1}{\sqrt{n}}\mathbf{y}\right\rangle_{\mathbb{R}^n}\right| \tag{38}$$

$$\leq \nu\|L_{m,R}^{-1}S_m\tilde{\phi}(t^*) - L_{m,R}^{-1}Sk(t^*,\cdot) + L_{m,R}^{-1}Sk(t^*,\cdot) - L_R^{-1}Sk(t^*,\cdot)\|_2 \tag{39}$$

$$\leq \nu\|L_{m,R}^{-1}(S_m\tilde{\phi}(t^*) - Sk(t^*,\cdot))\|_2 + \nu\|(L_{m,R}^{-1} - L_R^{-1})Sk(t^*,\cdot)\|_2 \tag{40}$$

$$\leq \nu/\gamma\|S_m\tilde{\phi}(t^*) - Sk(t^*,\cdot)\|_2 + \nu\|(L_{m,R}^{-1} - L_R^{-1})Sk(t^*,\cdot)\|_2. \tag{41}$$

Now, we can upper bound the two norms appearing in the expression above. The first addend can be directly bounded by applying Corollary C.3. The second addend in Equation (41) can be bounded by Lemma B.4. Hence, we obtain the following bound with probability at least $1-\delta$:

$$|\tilde{\mu}_{\text{post}}(t^*) - \mu_{\text{post}}(t^*)| \leq \nu/\gamma\|S_m\tilde{\phi}(t^*) - Sk(t^*,\cdot)\|_2 + \nu\|(L_{m,R}^{-1} - L_R^{-1})Sk(t^*,\cdot)\|_2 \tag{42}$$

$$\leq \sqrt{\frac{2\nu^2\alpha^4\log\frac{2Tn}{\delta}}{m\gamma^2}} + \frac{\sqrt{2}\kappa\nu}{\sqrt{\gamma}}\left[\frac{2\log\frac{8\kappa^2}{\gamma\delta}(1+\alpha^2/\gamma)}{3m} + \sqrt{\frac{2\log\frac{8\kappa^2}{\gamma\delta}\alpha^2}{\gamma m}}\right]. \tag{43}$$

The final result for the vector-valued GP can be obtained by applying a union bound.

## B.3 Proof of Theorem 4.5 (Deviation of Approximate Posterior Variance)

In this section, we prove our result related to the concentration of the approximate posterior variance. Again, we begin by stating some lemmas that will be used in the proof.

**Lemma B.5.** *Let $\delta = (0,1]$. Then, the following bound holds, with probability at least $1-\delta$,*

$$|\langle Sk(t^*,\cdot) - S_m\tilde{\phi}(t^*), L_R^{-1}Sk(t^*,\cdot)\rangle_{\mathbb{R}^n}| \leq \sqrt{\frac{2\kappa^2\alpha^4\log\frac{2Tn}{\delta}}{\gamma m}}. \tag{44}$$

*Proof.* By Cauchy-Schwarz,

$$|\langle Sk(t^*,\cdot) - S_m\tilde{\phi}(t^*), L_R^{-1}Sk(t^*,\cdot)\rangle_{\mathbb{R}^n}| \leq \|Sk(t^*,\cdot) - S_m\tilde{\phi}(t^*)\|_2\|L_R^{-1}Sk(t^*,\cdot)\|_2. \tag{45}$$

By using the polar decomposition of $S$, for a suitable partial isometry operator $U$, and according to Propositions B.1 and B.3

$$\|L_R^{-1} S k(t^*, \cdot)\|_2 \leq \|L_R^{-1}(SS^*)^{1/2} U\| \|k(t^*, \cdot)\|_{\mathcal{H}} \tag{46}$$

$$\leq \kappa \|L_R^{-1} L^{1/2}\| \tag{47}$$

$$\leq \kappa \|L_R^{-1/2}\| \|L_R^{-1/2} L^{1/2}\| \tag{48}$$

$$\leq \frac{\kappa}{\sqrt{\gamma}} \|L_R^{-1/2} L_\gamma^{1/2}\| \|L_\gamma^{-1/2} L^{1/2}\| \tag{49}$$

$$\leq \frac{\kappa}{\sqrt{\gamma}} \|L_\gamma^{-1/2} L_\gamma^{1/2}\| \tag{50}$$

$$\leq \frac{\kappa}{\sqrt{\gamma}}. \tag{51}$$

To conclude the proof, we can observe that, according to Corollary C.3,

$$\|S k(t^*, \cdot) - S_m \tilde{\phi}(t^*)\|_2 \leq \sqrt{\frac{2\alpha^4 \log \frac{2Tn}{\delta}}{m}}. \tag{52}$$

$\square$

**Lemma B.6.** *Let $\delta = (0, 1]$. Then, the following bound holds, with probability at least $1 - \delta$,*

$$|\langle S_m \tilde{\phi}(t^*), L_R^{-1}(S k(t^*, \cdot) - S_m \tilde{\phi}(t^*))\rangle_{\mathbb{R}^n}| \leq \frac{\alpha^2}{\gamma} \sqrt{\frac{2\alpha^4 \log \frac{2Tn}{\delta}}{m}}. \tag{53}$$

*Proof.* By Cauchy-Schwarz inequality and Proposition B.2,

$$|\langle S_m \tilde{\phi}(t^*), L_R^{-1}(S k(t^*, \cdot) - S_m \tilde{\phi}(t^*))\rangle_{\mathbb{R}^n}| \leq \|S_m \tilde{\phi}(t^*)\|_2 \|L_R^{-1}(S k(t^*, \cdot) - S_m \tilde{\phi}(t^*))\|_2 \tag{54}$$

$$\leq \|S_m\| \|\tilde{\phi}(t^*)\|_2 \|L_R^{-1}(S k(t^*, \cdot) - S_m \tilde{\phi}(t^*))\|_2 \tag{55}$$

$$\leq \frac{\alpha^2}{\gamma} \|S k(t^*, \cdot) - S_m \tilde{\phi}(t^*)\|_2. \tag{56}$$

Now, we can again observe that, according to Corollary C.3,

$$\|S k(t^*, \cdot) - S_m \tilde{\phi}(t^*)\|_2 \leq \sqrt{\frac{2\alpha^4 \log \frac{2Tn}{\delta}}{m}}, \tag{57}$$

which concludes the proof. $\square$

**Lemma B.7.** *Let $m \geq 8 \left(\frac{1}{3} + \frac{\alpha^2}{\gamma}\right) \log(\frac{8\alpha^2}{\gamma\delta})$, and $\delta = (0, 1]$. Then, the following bound holds, with probability at least $1 - \delta$,*

$$|\langle S_m \tilde{\phi}(t^*), (L_R^{-1} - L_{m,R}^{-1}) S_m \tilde{\phi}(t^*)\rangle_{\mathbb{R}^n}| \leq \frac{\alpha^3 \sqrt{2}}{\sqrt{\gamma}} \left[ \frac{2 \log \frac{8\kappa^2}{\gamma\delta}(1 + \alpha^2/\gamma)}{3m} + \sqrt{\frac{2 \log \frac{8\kappa^2}{\gamma\delta} \alpha^2}{\gamma m}} \right]. \tag{58}$$

*Proof.* Firstly, we can observe that, by Cauchy-Schwarz inequality, Propositions B.2 and B.3, the polar decomposition of $S_m$, and the fact that $L_m \preccurlyeq L_{m,\gamma}$ by definition, we have that

$$|\langle S_m \tilde{\phi}(t^*), (L_R^{-1} - L_{m,R}^{-1}) S_m \tilde{\phi}(t^*)\rangle_{\mathbb{R}^n}|$$

$$= |\langle S_m \tilde{\phi}(t^*), L_{m,R}^{-1}(L - L_m) L_R^{-1} S_m \tilde{\phi}(t^*)\rangle_{\mathbb{R}^n}| \tag{59}$$

$$= |\langle L_{m,R}^{-1/2} S_m \tilde{\phi}(t^*), L_{m,R}^{-1/2}(L - L_m) L_R^{-1} S_m \tilde{\phi}(t^*)\rangle_{\mathbb{R}^n}| \tag{60}$$

$$\leq \|L_{m,R}^{-1/2} S_m \tilde{\phi}(t^*)\|_2 \|L_{m,R}^{-1/2}(L - L_m) L_R^{-1} S_m \tilde{\phi}(t^*)\|_2 \tag{61}$$

$$\leq \alpha^3 \|L_{m,R}^{-1/2}(S_m S_m^*)^{1/2} U\| \|L_{m,R}^{-1/2} L_{m,\gamma}^{1/2} L_{m,\gamma}^{-1/2}(L - L_m) L_R^{-1} S_m \tilde{\phi}(t^*)\|_2 \tag{62}$$

$$\leq \alpha^3 \|L_{m,R}^{-1/2} L_{m,\gamma}^{1/2}\| \|L_{m,\gamma}^{-1/2} L_\gamma^{1/2}\| \|L_\gamma^{-1/2}(L - L_m) L_\gamma^{-1/2}\| \|L_\gamma^{1/2} L_R^{-1/2}\| \|L_R^{-1/2}\| \tag{63}$$

$$\leq \frac{\alpha^3}{\sqrt{\gamma}} \|L_{m,\gamma}^{-1/2} L_\gamma^{1/2}\| \|L_\gamma^{-1/2}(L - L_m) L_\gamma^{-1/2}\|. \tag{64}$$

Now, we can bound the two factors. According to Propositions B.1 and C.4, with probability at least $1 - \delta$, for $\delta \in (0, 1]$ and $m \geq 8\left(\frac{1}{3} + \frac{\alpha^2}{\gamma}\right) \log(\frac{8\alpha^2}{\gamma\delta})$, we have that

$$\|L_{m,\gamma}^{-1/2} L_\gamma^{1/2}\| \|L_\gamma^{-1/2}(L_\gamma - L_{m,\gamma}) L_\gamma^{-1/2}\| \leq \sqrt{2}\left[\frac{2\log\frac{8\kappa^2}{\gamma\delta}(1 + \alpha^2/\gamma)}{3m} + \sqrt{\frac{2\log\frac{8\kappa^2}{\gamma\delta}\alpha^2}{\gamma m}}\right], \tag{65}$$

concluding the proof. $\qquad\square$

**Proof of Theorem 4.5** We are now ready to prove Theorem 4.5. According to Equations (8) and (10), and similarly to what we did for the posterior mean, we can decompose the error on the variance of a single DOF as follows:

$$\begin{aligned}
|\sigma_{\text{post}}^2(t^*) - \tilde{\sigma}_{\text{post}}^2(t^*)| = &|k(t^*, t^*) - \langle Sk(t^*, \cdot), L_R^{-1} Sk(t^*, \cdot)\rangle_{\mathbb{R}^n} \\
&- \tilde{k}(t^*, t^*) + \langle S_m \tilde{\phi}(t^*), L_{m,R}^{-1} S_m \tilde{\phi}(t^*)\rangle_{\mathbb{R}^n}| \tag{66} \\
\leq &|k(t^*, t^*) - \tilde{k}(t^*, t^*)| \\
&+ |\langle Sk(t^*, \cdot), L_R^{-1} Sk(t^*, \cdot)\rangle_{\mathbb{R}^n} - \langle S_m \tilde{\phi}(t^*), L_{m,R}^{-1} S_m \tilde{\phi}(t^*)\rangle_{\mathbb{R}^n}| \tag{67} \\
\leq &|k(t^*, t^*) - \tilde{k}(t^*, t^*)| \\
&+ |\langle Sk(t^*, \cdot) - S_m \tilde{\phi}(t^*), L_R^{-1} Sk(t^*, \cdot)\rangle_{\mathbb{R}^n}| \\
&+ |\langle S_m \tilde{\phi}(t^*), L_R^{-1} Sk(t^*, \cdot) - L_{m,R}^{-1} S_m \tilde{\phi}(t^*)\rangle_{\mathbb{R}^n}| \tag{68} \\
\leq &|k(t^*, t^*) - \tilde{k}(t^*, t^*)| \\
&+ |\langle Sk(t^*, \cdot) - S_m \tilde{\phi}(t^*), L_R^{-1} Sk(t^*, \cdot)\rangle_{\mathbb{R}^n}| \\
&+ |\langle S_m \tilde{\phi}(t^*), L_R^{-1}(Sk(t^*, \cdot) - S_m \tilde{\phi}(t^*))\rangle_{\mathbb{R}^n}| \\
&+ |\langle S_m \tilde{\phi}(t^*), (L_R^{-1} - L_{m,R}^{-1}) S_m \tilde{\phi}(t^*)\rangle_{\mathbb{R}^n}|. \tag{69}
\end{aligned}$$

Now, we can upper bound the four addends appearing in the decomposition above. The first addend can by directly bounded by Corollary C.2. The second addend of the decomposition in Equation (69) can be bounded by Lemma B.5. The third addend in Equation (69) can be bounded by Lemma B.6. The last addend in Equation (69) can be bounded by Lemma B.7. In this way, we retrieve the result of Theorem 4.5, obtaining the following bound holding with probability at least $1 - \delta$. Having defined

$$C := \sqrt{\frac{2\alpha^4 \log \frac{2T}{\delta}}{m}} + \sqrt{\frac{2\kappa^2 \alpha^4 \log \frac{2Tn}{\delta}}{\gamma m}} + \frac{\alpha^2}{\gamma}\sqrt{\frac{2\alpha^4 \log \frac{2Tn}{\delta}}{m}} + \frac{\alpha^3 \sqrt{2}}{\sqrt{\gamma}}\left[\frac{2\log \frac{8\kappa^2}{\gamma\delta}(1+\alpha^2/\gamma)}{3m} + \sqrt{\frac{2\log \frac{8\kappa^2}{\gamma\delta}\alpha^2}{\gamma m}}\right]:$$

$$
\begin{aligned}
|\sigma^2_{\text{post}}(t^*) - \sigma^2_{\text{post}}(t^*)| &\leq |\langle k(t^*,\cdot), k(t^*,\cdot)\rangle_{\mathcal{H}} - \langle \tilde{\phi}(t^*), \tilde{\phi}(t^*)\rangle_{\mathbb{R}^m}| \\
&\quad + \langle Sk(t^*,\cdot) - S_m\tilde{\phi}(t^*), L_R^{-1}Sk(t^*,\cdot)\rangle_{\mathbb{R}^n}| \\
&\quad + |\langle S_m\tilde{\phi}(t^*), L_R^{-1}(Sk(t^*,\cdot) - S_m\tilde{\phi}(t^*))\rangle_{\mathbb{R}^n}| \\
&\quad + |\langle S_m\tilde{\phi}(t^*), (L_R^{-1} - L_{m,R}^{-1})S_m\tilde{\phi}(t^*)\rangle_{\mathbb{R}^n}| \quad\quad\quad (70) \\
&\leq C. \quad\quad\quad\quad\quad\quad\quad\quad\quad\quad\quad\quad\quad\quad\quad\quad\quad\quad (71)
\end{aligned}
$$

The final result for the vector-valued GP can be obtained by applying a union bound.

## C  Concentration Results

We first provide a few lemmas for the concentration of the approximate kernel functions that derive from Hoeffding inequality, and then a lemma for the concentration of random operators that derives from Bernstein inequality. Again, we denote by $A_\gamma$ the operator $A + \gamma I$. $\|\cdot\|$ denotes the operator norm, while $\|\cdot\|_2$ denotes the Euclidean norm of a vector.

### C.1  Approximation of the Kernel Function

Note that if a *uniform* convergence of the RF-HGP posterior is sought w.r.t. the domain of the function modelled with the HGP, our proofs could be adapted by replacing the following Corollary C.2 with a uniform convergence result. For instance, in the case of RFFs, such a result can be found in [39, Theorem 1 and Remark 1].

**Lemma C.1.** *Let $\delta = (0,1]$. Then, for any $(t_1, t_2)$, with probability at least $1 - \delta$, it holds*

$$\left|\tilde{\phi}(t_1)^T\tilde{\phi}(t_2) - k(t_1, t_2)\right| \leq \sqrt{\frac{2\alpha^4 \log \frac{2}{\delta}}{m}}. \quad\quad\quad (72)$$

*Proof.* To upper bound the quantity of interest, we can use Hoeffding's inequality for bounded random variables. Let $A_j(t_1, t_2) := \psi_j(t_1)\psi_j(t_2) - \mathbb{E}_\omega \psi(\omega, t_1)\psi(\omega, t_2)$. Since $-\alpha^2 \leq \psi_j(t_1)\psi_j(t_2) \leq \alpha^2$ according to Assumption 4.2, by Hoeffding inequality, we have that

$$\Pr\left\{\frac{1}{m}\left|\sum_{j=1}^m A_j(t_1, t_2)\right| \geq \frac{t}{m}\right\} \leq 2e^{-\frac{2t^2}{4m\alpha^4}}. \quad\quad\quad (73)$$

Therefore, by setting the above upper bound smaller than $\delta$, for $\delta \in (0,1]$, we get that with probability at least $1 - \delta$

$$\left|\tilde{\phi}(t_1)^T\tilde{\phi}(t_2) - k(t_1, t_2)\right| = \frac{1}{m}\left|\sum_{j=1}^m A_j(t_1, t_2)\right| \leq \sqrt{\frac{2\alpha^4 \log \frac{2}{\delta}}{m}}. \quad\quad\quad (74)$$

$\square$

**Corollary C.2.** *Let $\delta = (0,1]$. Then with probability at least $1 - \delta$, it holds*

$$\left|\tilde{\phi}(t^*)^T\tilde{\phi}(t^*) - k(t^*, t^*)\right| \leq \sqrt{\frac{2\alpha^4 \log \frac{2|\mathcal{T}|}{\delta}}{m}}, \quad \forall t^* \in \mathcal{T}. \quad\quad\quad (75)$$

*Proof.* We apply Lemma C.1 on each element of $\mathcal{T}$ with $\delta' := \delta/T$. The claimed result then follows using a union bound. $\square$

**Corollary C.3.** *Let* $\delta = (0, 1]$. *Then with probability at least* $1 - \delta$,

$$\|S_m \tilde{\phi}(t^*) - Sk(t^*, \cdot)\|_2 \leq \sqrt{\frac{2\alpha^4 \log \frac{2|\mathcal{T}|n}{\delta}}{m}}, \quad \forall t^* \in \mathcal{T}. \tag{76}$$

*Proof.* It holds

$$\|S_m \tilde{\phi}(t^*) - Sk(t^*, \cdot)\|_2^2 = \frac{1}{n} \sum_{i=1}^{n} \left[ \tilde{\phi}(t_i)^T \tilde{\phi}(t^*) - k(t_i, t^*) \right]^2 \tag{77}$$

The result thus follows from applying $nT$ times Lemma C.1 on the pairs $((t_i, t^*))_{1 \leq i \leq n, t^* \in \mathcal{T}}$ with probability $\delta' := \delta/(nd)$ and using a union bound. $\square$

## C.2 Concentration of the Kernel matrix

The following result derives from the Bernstein inequality for sums of random operators on separable Hilbert spaces in operator norm.

**Proposition C.4** (Proposition 6 and Remark 10 of [20]). *Let* $\mathbf{v}_1, ..., \mathbf{v}_p$ *with* $p \geq 1$, *be independent and identically distributed random vectors on a separable Hilbert spaces* $\mathcal{H}$ *such that* $Q = \mathbb{E}\mathbf{v} \otimes \mathbf{v}$ *is trace-class, and for any* $\lambda > 0$ *there exists a constant* $\mathcal{F}_\infty(\lambda) < \infty$ *such that* $\langle \mathbf{v}, (Q + \lambda I)^{-1} \mathbf{v} \rangle \leq \mathcal{F}_\infty(\lambda)$ *almost everywhere. Let* $Q_p = \frac{1}{p} \sum_{i=1}^{p} \mathbf{v}_i \otimes \mathbf{v}_i$ *and take* $0 < \lambda \leq \|Q\|$. *Then for any* $\delta \geq 0$, *the following holds with probability at least* $1 - \delta$:

$$\|Q_\lambda^{-1/2}(Q - Q_p)Q_\lambda^{-1/2}\| \leq \frac{2w(1 + \mathcal{F}_\infty(\lambda))}{3p} + \sqrt{\frac{2w\mathcal{F}_\infty(\lambda)}{p}} \tag{78}$$

*where* $w = \log \frac{8 \operatorname{Tr} Q}{\lambda \delta}$. *Moreover, with the same probability,*

$$\lambda_{max}\left[ Q_\lambda^{-1/2}(Q - Q_p)Q_\lambda^{-1/2} \right] \leq \frac{2w}{3p} + \sqrt{\frac{2w\mathcal{F}_\infty(\lambda)}{p}}. \tag{79}$$

*Moreover, for any* $s \in (0, 1]$, *if* $\|\mathbf{v}_i\| \leq \alpha$, *we have that, with probability at least* $1 - \delta$,

$$\lambda_{max}\left[ Q_\lambda^{-1/2}(Q - Q_p)Q_\lambda^{-1/2} \right] \leq s. \tag{80}$$

*provided that* $p \geq \frac{2}{t^2}\left[ \frac{2t}{3} + \mathcal{F}_\infty(\gamma) \right] \log \frac{8\alpha^2}{\lambda \delta}$ *and* $\lambda \leq \|Q\|$.

# D Efficient Matrix Inversion and Online Updates

In this section, we show how the expression of the posterior mean and variance can easily be updated when adding new samples to the dataset.

We recall that the operators $S_m$ and $L_{m,R}$ are matrices and are defined in Appendix A. As discussed in Appendix A, the inversion of $L_{m,R}$, involved the posteriors of Equations (9) and (10), can be simplified by applying Woodbury identity [36], as follows:

$$L_{m,R}^{-1} = (S_m S_m^* + R)^{-1} \tag{81}$$

$$= R^{-1} - R^{-1}S_m(I + S_m^* R^{-1} S_m)^{-1} S_m^* R^{-1}. \tag{82}$$

The posterior mean of the HGP in Equation (9) becomes:

$$\tilde{\mu}_{\text{post}}(t^*) = \langle \left[ R^{-1} - R^{-1}S_m(I + S_m^* R^{-1} S_m)^{-1} S_m^* R^{-1} \right] S_m \tilde{\phi}(t^*), \frac{1}{\sqrt{n}}\mathbf{y} \rangle_{\mathbb{R}^n} \tag{83}$$

$$= \tilde{\phi}(t^*)^T \left[ I - S_m^* R^{-1} S_m(I + S_m^* R^{-1} S_m)^{-1} \right] S_m^* R^{-1} \frac{1}{\sqrt{n}}\mathbf{y} \tag{84}$$

$$= \tilde{\phi}(t^*)^T \left[ I - B(I + B)^{-1} \right] A \tag{85}$$

where $A := \frac{1}{\sqrt{n}} S_m^* R^{-1} \mathbf{y} \in \mathbb{R}^m$ and $B := S_m^* R^{-1} S_m \in \mathbb{R}^{m \times m}$. Moreover, the only term in the expression of the posterior variance of Equation (10)

$$\tilde{\sigma}^2_{\text{post}}(t^*) = \langle \tilde{\phi}(t^*), \tilde{\phi}(t^*) \rangle_{\mathbb{R}^m} + n r_{t^*} - \langle S_m \tilde{\phi}(t^*), (L_m + R)^{-1} S_m \tilde{\phi}(t^*) \rangle_{\mathbb{R}^n} \tag{86}$$

which varies with $n$ is

$$\langle S_m \tilde{\phi}(t^*), (L_m + R)^{-1} S_m \tilde{\phi}(t^*) \rangle_{\mathbb{R}^n} \tag{87}$$

$$= \tilde{\phi}^T(t^*) \left[ S_m^* R^{-1} S_m - S_m^* R^{-1} S_m (I + S_m^* R^{-1} S_m)^{-1} S_m^* R^{-1} S_m \right] \tilde{\phi}(t^*), \tag{88}$$

$$= \tilde{\phi}^T(t^*) \left[ B - B(I + B)^{-1} B \right] \tilde{\phi}(t^*). \tag{89}$$

When a new human demonstration is gathered, the training set is enlarged by adding $n_{new}$ training points. This means that the matrix $S_m$ is updated by adding $n_{new}$ rows (and renormalized), containing the RF embeddings of the new training points. The same happens to vector $\mathbf{y}$ and to the diagonal matrix $R$, which is enlarged by adding $n_{new}$ rows and columns. This means that matrices $A$ and $B$ support online updates. In particular, after initializing $A$ and $B$ to the null matrix, having collected the new embeddings in $S_{m,new} \in \mathbb{R}^{n_{new} \times m}$ (with normalization $n_{new}^{-1/2}$) and the new noise variance values in $R_{new} \in \mathbb{R}^{n_{new} \times n_{new}}$ (with normalization $n_{new}^{-1}$), the updates are as follows:

$$A \leftarrow A + \frac{1}{\sqrt{n_{new}}} S_{m,new}^* R_{new}^{-1} \mathbf{y}_{new} \tag{90}$$

$$B \leftarrow B + S_{m,new}^* R_{new}^{-1} S_{m,new}. \tag{91}$$

Having computed the updates, the matrices appearing in the posterior mean and variance can be computed in constant time w.r.t. the current size of the training set during the data streaming.

