# OpenReview forum: "Heteroscedastic Gaussian Processes and Random Features: Scalable Motion Primitives with Guarantees"
_robot-learning.org/CoRL/2023/Conference — CoRL 2023 Poster_

### Official Review · Reviewer_JwFW · 2023-06-26

**Confidence:** 5
**Originality:** Fair
**Technical Quality:** Good
**Clarity Of Presentation:** Very Good
**Impact:** 2

**Recommendation:**

Weak Reject: I recommend rejecting the paper, but will not argue for my recommendation if the majority of other reviewers have a different opinion.

**Review:**

The main strength of the paper is the theoretical analysis leading to the bounds for the approximation error of HGP with RF approximation. The main interest of these bounds is that they apply to the heteroscedastic case. In addition, the paper is clearly written.

Despite the novel bound, I am generally concerned with the originality and significance of the paper, as well as with the application scope in which the contributions are phrased.

First, HGPs have been widely studied and already applied to robot learning, see e.g., [4, 5]. Second, a large body of work is concerned with reducing the computational cost of (H)GPs with respect to the number of training samples, including, as discussed in the paper, variational methods based on inducing points and feature-based kernel approximations. Therefore, the RF approximation considered in the paper is not new and has been widely studied in the literature. In addition, recent works such as [Hensman, 2018], [Dutordoir, 2020], [Cunningham, 2023] (to cite a few) proposed efficient variational and/or feature-based approximations beyond RF approximations allowing the consideration of millions of datapoints with some order of magnitude faster than traditional (sparse) GPs. In other words, there exist other, more performant approximations that considerably reduce the computational cost of GPs and that could be considered in robot learning as well.

Concerning the scope, the paper seems to limit the input space of the considered GPs to time (i.e., R), while GP are generally interesting because they are able to deal with high-dimensional input spaces. I am not sure if the main theorems 4.4 and 4.5 depend on this assumption. If not, it would be good to precise it. Moreover, the paper phrases its main contributions in the context of variable impedance learning (VIL). It seems to me that HGPs with RF approximations are applicable to many other robotic problems, including, e.g., learning from demonstrations of general trajectories, Bayesian optimization, etc. Therefore, reducing the scope of the paper to VIL generally undermines the contributions of the paper.

Finally, the experimental evaluation validates the fact that RF approximations are computationally cheaper than computing the HGP without approximation and that they provide a good approximation of the mean and variance. However, I would have liked to see a clear link between the empirical evaluation and the bounds provided in Section 4. For instance, the bound may be represented in Figures 3 and 6. Moreover, the performance of the RF approximation is not evaluated against other efficient variational and/or feature-based approximations. This would provide the reader with evident to choose one of these approximation when using GPs or HGPs in the context of robot learning from demonstrations. In particular, a comparison against classical variational GPs would be straightforward. Moreover, as recent approaches have been shown to be very efficient to approximate GPs with many training points, they should be explicitly considered in the paper.


[Hensman, 2018] Hensman, Durrande, & Solin, "Variational Fourier Features for Gaussian Processes", JMLR, 2018.

[Dutordoir, 2020] Dutordoir, Durrande, & Hensman, "Sparse Gaussian Processes with Spherical Harmonic Features", ICML, 2020.

[Cunningham, 2023] Cunningham, Augusto de Souza, Takao, van der Wilk, & Deisenroth, "Actually Sparse Variational Gaussian Processes", AISTATS, 2023.


--------
Post-rebuttal comments:

Although I appreciate the efforts of the authors to answer the reviewers during the rebuttal, my general concerns about the paper mostly remain.

The main contribution of the paper is the presented theoretical analysis leading to the bounds for the approximation error of HGP with RF approximation. However, the contributions of the paper remain very limited from a robotics perspective as (i) HGPs have been widely studied and already applied to robot learning, and (ii) the paper shows only a single case of application (using the tasks described in [10]) where the derived bound may be useful. Despite that I appreciate the presented theoretical derivation, I find that its impact on (robotics) applications is not well reflected by the paper. In this sense, I agree with Reviewer qoam that the theoretical contributions of the paper may be better suited for publication within the ML / GP community due to its potential applications beyond the robotics domain.
If there is interest to publish the paper for robotics, I would expect to see a clear discussion on the impact of its results for robotics applications.

Lastly, I appreciate the discussion provided during the rebuttal about RF approximations and variational approaches. However, I believe that the newly-added comparison lacks significance, and I think that the discussion (and eventual comparisons) should consider more recent works and implementations.

**Quality Of The Limitations Section:**

Additional details required

**Questions For Rebuttal:**

- How does the considered RF approximation related to recent efficient variational and/or feature-based approximations in terms of computational cost and approximation error?
- Are Theorems 4.4 and 4.5 limited to input domain in R or are they valid for any input domain?

**Robotics Focus:**

Sufficient demonstration on hardware

**Summary Of Paper:**

This paper provides bounds for the approximation error for the mean and variance of heteroscedastic Gaussian processes (HGs) whose kernel is approximated via random features (RFs). The interest of such approximation is to reduce the computational complexity of the posterior computation by reducing the cost of the training kernel matrix inversion. The resulting HGP are evaluated in the context of variable impedance learning from demonstrations.

**Summary Of Recommendation:**

Although the provided bounds seems theoretically sound, I believe that the paper generally lacks significance and novelty. Moreover, it does not consider the latest state of the art in terms of efficient GPs and its experimental evaluation is limited. Therefore, I recommend rejecting the paper.

----
Post-rebuttal:

I updated my score from strong to weak reject in appreciation for the theoretical bound presented in Section 4. However, I still advocate for rejection as the rest of the paper generally lacks novelty, and the impact of Section 4 in (robotics) applications beyond VIL remains unclear.

---

### Official Review · Reviewer_qoam · 2023-07-09

**Confidence:** 4
**Originality:** Good
**Technical Quality:** Very Good
**Clarity Of Presentation:** Very Good
**Impact:** 3

**Recommendation:**

Weak Accept: I recommend accepting the paper, but will not argue for my recommendation if the majority of other reviewers have a different opinion.

**Review:**

The use of GPs for regularized learning of demonstrations enjoys a long history, and some rigorous theoretical analysis is welcome.
When reading the paper, it sometimes felt it could be better suited as a ML paper on heteroscedastic GPs, as it's never made clear what makes learning from demonstrations special as a heteroscedastic regression problem. Usually in movement primitive research there is a focus on incorporating different inductive biases, such as periodicity, and blending lower-level primitives through composability.

**Why are demonstrations heteroscedastic?** The central heteroscedasticity assumption here is motivated by the demonstration data / expert being inconsistent and 'corrupted'. However, while there is some motor noise at play, the inconsistency is often more due to ambiguity. For example, for a 2D go-to task, the expert trajectory could be a straight line, or an arc above or below it. If you fit this data into a heteroscedastic model and sample predictions IID, you will not recover these trajectories samples but rather rough heteroscedastic noise. It could be argued that the demonstration ambiguity is a form of permanent task-induced epistemic uncertainty, rather than aleatoric uncertainty. The benefit of fitting the predictive distribution of a single GP over the whole trajectory sequence (which I discuss in the question section) is that sampling the model in function space should yield trajectories like the demonstrations. Since this model has two GPs for the mean and variance, sampling both in function space and combining should yield the correlated trajectories one would expect? This feature is important (and perhaps unique to LfD) because if these learned primitives are used to initialize an RL finetuning step, you want to sample trajectories that explore 'correctly' by reproducing the correlations of the demonstrations. I think a discussion regarding whether we are modelling aleatoric or epistemic uncertainty, as well as the function-space view of the trajectories data, is a useful and necessary addition.

**Scholarship.** The authors don't cite many robotics papers and omit discussing many similar prior works on the intersection of GPs and movement primitives, for example

Probabilistic movement primitives / Using Probabilistic Movement Primitives in Robotics
A Paraschos, C Daniel, J Peters NeurIPS 2013 / AuRo

Kernelized movement primitives
Y Huang et al IJRR 2019

Probabilistic movement primitives for coordination of multiple human–robot collaborative tasks
Maeda et al. Autonomous robots 2017

Latent spaces for dynamic movement primitives
S Bitzer, S Vijayakuma ICRA 2009

Learning from demonstration with model-based Gaussian process
N Jaquier et al Conference on Robot Learning 2020

Differentiable Gaussian Process Motion Planning
Bhardwaj et al ICRA 2020

Adaptation and Robust Learning of Probabilistic Movement Primitives
Gomez-Gonzalez et al. TRO 2020

Bayesian Disturbance Injection: Robust Imitation Learning of Flexible Policies
Oh et al, ICRA 2021

ProDMP: A Unified Perspective on Dynamic and Probabilistic Movement Primitives
Li et al RAL IEEE 2023

There was also a paper at CoRL last year using GPs as action priors for MPC and looked at random Fourier features

Inferring Smooth Control: Monte Carlo Posterior Policy Iteration with Gaussian Processes
J Watson, J Peters - Conference on Robot Learning, 2023

While probabilistic movement primitives are not sold as GPs, if you look at the math they are essentially Bayesian linear regression models with RBF features, so almost like a sparse inducing point GP approximation.  The difference is that the weight distribution is fit in weight-space, by fitting a primitives per demonstration first, rather than prediction space over all demonstrations. This is in order to not make the IID assumption.

**Quadrature Random features.** Lines 52 - 55 seems to separate QRFFs from RFFs, but QRFFs are using Gauss Hermite quadrature to approximate the Gaussian integral that RFFs use Monte Carlo samples for. Whats nice about QRFFs is that when the input is 1 dimensional (like time in this paper) the curse of dimensionality associated with Gauss Hermite quadrature is not so bad and you can achieve a very good approximation of the integral. The authors are correct that this approximation truncates the spectrum, but this is true for RFFs too in practice based on whatever the largest frequency sample is.


**Quality Of The Limitations Section:**

Limitations are addressed clearly

**Questions For Rebuttal:**

1. Since you have a Bayesian linear regression model with the random features, could it not be easier to just fit the predictive distribution over trajectories directly rather than the EM approach? E.g.

Parametric Gaussian Process Regressors
Martin Jankowiak, Geoff Pleiss, Jacob Gardner ICML 2020

It would be rather high-dimensional, but the training would be a lot simpler and more inline with behaviour cloning.

2. See earlier question about function space sampling recovering the temporal correlations of the demonstrations.

3. Why aren't inducing points used as a baseline? Line 285 says they are more complex, but I think the inducing points in the time domain have a nice intuitive interpretation as via-points .

4. Could it be beneficial to use both RFFs and RBF features for the approximation? Since RBFs are good at modelling the datapoints. C.f.

Efficiently sampling functions from Gaussian process posteriors
James Wilson, Viacheslav Borovitskiy, Alexander Terenin, Peter Mostowsky, Marc Deisenroth, ICML 2020

**Robotics Focus:**

Sufficient demonstration on hardware

**Summary Of Paper:**

The authors motivate heteroscedastic Gaussian process regression for learning from demonstrations, with random Fourier features to approximate the kernel.

**Summary Of Recommendation:**

I advocate for acceptance based on the strong theoretical contribution, robot experiments and relevance to CoRL.

---

### Official Review · Reviewer_zk8k · 2023-07-18

**Confidence:** 1
**Originality:** Good
**Technical Quality:** Good
**Clarity Of Presentation:** Fair
**Impact:** 2

**Recommendation:**

Weak Accept: I recommend accepting the paper, but will not argue for my recommendation if the majority of other reviewers have a different opinion.

**Review:**

I didn’t find the paper to very well organized even though the individual sections were clearer to read. The was not a clear section on Related Works (some subsections of Section 4 seemed to refer to related approaches, but works related to ). Furthermore, the paper covers several different concepts – learning a motion primitive, learning a heteroscedastic GP with RFF approximation and bounds on this approximation -  that are not presented cohesively. The presented approximation bounds for RFF seemed significant. But it seems to be a strict improvement over previously presented bounds under some assumptions that seem to be too conservative.


**Strengths:**
- theoretical results
- results on hardware

**Weaknesses:**
- Paper didn’t seem to be organized very well
- Some clarity issues: for example it is not clear to me what ’d’ is. Is it the dimensionality of the outputs?
- There isn't much reporting on related works. This work is not compared to other approaches such as probabilistic movement primitives and other dynamic movement primitives that can also be used for LfD.


**Quality Of The Limitations Section:**

Limitations are addressed clearly

**Questions For Rebuttal:**

1. What is 'd'? If 'd' is the output dimensionality, it seems that d>1 in the experiments, and the strict improvement in the error bounds does not hold.

**Robotics Focus:**

Sufficient demonstration on hardware

**Summary Of Paper:**

This paper learns a vector-values heteroscedastic GP using Random Fourier features (RFF) for modeling motion primitives. The authors additionally provide bounds on the approximation error of the posterior mean and variance of the HGP approximated through  RFF .

**Summary Of Recommendation:**

I did not find the paper clear and cohesive. The motivation of the paper was especially not clear to me.

-----------POST REBUTTAL ------------

Based on the rebuttal provided by the authors, I am happy to raise my score to accept

---

### Official Review · Reviewer_kgu9 · 2023-07-21

**Confidence:** 4
**Originality:** Good
**Technical Quality:** Very Good
**Clarity Of Presentation:** Very Good
**Impact:** 4

**Recommendation:**

Weak Accept: I recommend accepting the paper, but will not argue for my recommendation if the majority of other reviewers have a different opinion.

**Review:**

The main contribution of the paper seems to be the theoretical results in Sec. 4, which bound the approximation error of the posterior mean and variance of the proposed model with respect to the posterior of an exact GP model. I haven't checked the proofs, but the results seem to be novel and are should be beneficial to applications beyond the robotics domain. However, I have a few concerns regarding a few claims in the paper and the experimental results, which I detail in my questions.

**Quality Of The Limitations Section:**

Limitations are addressed clearly

**Questions For Rebuttal:**

1. Sec. 4.2, **Feature approximations.** The error bounds in Mutny and Krause [16] deal with possibly continuous function domains and data that might be not i.i.d. The results in Thr. 4.4 and 4.5, however, hold for a finite domain $\mathcal{T}$, and the size of the domain $|\mathcal{T}|$ is a component of the error bound. So I am not sure if the claimed improvement in convergence rate is applicable.

2. Sec. 4.2, **convergence rate.** I could not see how a convergence rate of $\mathcal{O}(n^{3/2}\log n)$ comes out of the results in Thr. 4.4 and 4.5, when the only terms explicitly dependent on $n$ are $\mathcal{O}(\sqrt{\log n})$.

3. **Experimental comparisons.** There are no experimental comparisons against other sparse GP approximations that can deal with heteroscedastic noise, such as variational approximations based on inducing points. The only baseline is an exact GP. I believe comparisons against another sparse GP baseline should strengthen the paper and provide better insight into the significance of it contribution, especially if there are gains with respect to runtime complexity or approximation errors.

4. **Application to learning from demonstration.** The paper also seems not to apply the proposed RF-HGP model to the generation of motion primitives within robotic control or motion planning tasks. So it is hard to assess the impact of such modelling framework within the robotic tasks it has been proposed for.

Minor issues:
* When dealing with sparse spectrum GPs, we usually do not resort to eq. 4 and 5 to compute the posterior mean and variance, but to alternative versions obtained by the Woodbury identities, as mentioned in the paper. I believe it would be useful to have such versions explicitly available somewhere in the main paper.

* What are the total number of data points for each task in the experiments? I've only found mention that they are in the order of $10^3$, but not an explicit number.

**Robotics Focus:**

Sufficient demonstration on hardware

**Summary Of Paper:**

This paper proposes a sparse spectrum Gaussian process model for settings with heteroscedastic noise in learning from demonstration and provides theoretical guarantees for the approximation error with respect to an exact Gaussian process (GP) model. The analysis is based on previous results in the literature on random Fourier features, especially Rudi and Rosasco [25]. An algorithm is also provided to learn the hyper-parameters of the model via expectation maximisation, adapting the method proposed by Kersting et al. [4]. Experimental results with data collected from tasks executed with a robotic arm show the evolution of the error in the approximation and the computation time with respect to the number of Fourier features.

**Summary Of Recommendation:**

My concerns are mainly due some claims in the theoretical analysis and issues with the experimental comparisons. I'll be looking forward to clarifications during the author discussion phase.

---

### Author Response · Authors · 2023-08-10
**Common answer**

We answer here to the concerns raised by multiple reviewers, and then provide a separated rebuttal for each reviewer. A pdf document with new experimental results is attached to each individual rebuttal (as it cannot be attached here).

**Comparison with variational approaches**

We agree that inducing point approximations could be a valuable baseline. Some variational algorithms, such as SVGP by Hensman et al. (2013), are particularly suitable for large-scale learning, and for this reason we supply a comparison with this algorithm. In general, we observe that SVGP requires many more iterations to converge, in the hyperparameter training, compared to the RF-HGP, due to the greater complexity of the optimization problem to be solved. To overcome this issue, we set the maximum number of iterations to 100. However, as we can observe from the plots in the attached figure, this hinders the convergence through the EM training process, and our proposed approach has a strictly better accuracy-computational complexity tradeoff. We will refactor our heuristic experiments to include this comparison.

---

### Decision · Program_Chairs · 2023-08-30

**Decision:**

Accept (Poster)

**Comment:**

The paper combines Gaussian processes with random Fourier features to derive a variable observational noise model (aka heteroskedastic) used for learning from demonstration. Theoretical guarantees for the approximation are provided as well as an algorithm to learn the hyperparameters of the model. Experiments with a robotic arm demonstrate the benefits of the approach on assembly and bed-making tasks.

Reviewers were positive about the theoretical contributions of the work but were concerned about the presentation and the overall connection and relevance of these results to robotics. In the rebuttal and in the new version, authors have addressed most of these issues but the significance to robotics remains unclear. All in all, I believe the theoretical contributions are important and can be applied to other problems authors have yet to consider.